REGISTERED REPORT PROTOCOL

# Hypocalcaemia and calcium intake in pregnancy: A research protocol for critical analysis of risk factors, maternofoetal outcomes and evaluation of diagnostic methods in a third-category health facility, Cameroon

Atem Bethel Ajong[1,2☯]*, Bruno Kenfack[3☯], Innocent Mbulli Ali[2‡], Martin Ndinakie Yakum[4‡], Loai Aljerf[5‡], Phelix Bruno Telefo[2☯]

1 Kekem District Hospital, Kekem, West Region, Cameroon, 2 Department of Biochemistry, University of Dschang, Dschang, West Region, Cameroon, 3 Department of Obstetrics / Gynaecology and Maternal Health, Faculty of Medicine and Pharmaceutical Sciences, University of Dschang, Dschang, West Region, Cameroon, 4 Medecins Sans Frontieres-Spain (MSF-OCBA), Maiduguri, Nigeria, 5 Faculty of Dentistry, Damascus University, Damascus, Syria

☯ These authors contributed equally to this work.
‡ These authors also contributed equally to this work.
* christrah@yahoo.fr

This is a Registered Report and may have an associated publication; please check the article page on the journal site for any related articles.

## Abstract

### Introduction

Hypocalcaemia in pregnancy remains a major health issue, particularly in the developing world where daily calcium intakes are suboptimal. This electrolyte imbalance can lead to severe maternofoetal and childhood consequences. Calcium supplementation, amongst others, contributes significantly to meeting calcium demands in pregnancy. With ionised calcaemia as the gold standard for diagnosis, total calcaemia and albumin-corrected calcaemia in other pathological states have been found to overestimate the burden of hypocalcaemia. The main objectives of this study are to describe the blood calcium level (total, albumin corrected, and ionised calcaemia) and associated maternofoetal outcomes while identifying determinants of calcium supplementation and ionised hypocalcaemia. This study will also evaluate the sensitivity and specificity of albumin corrected calcaemia as a diagnostic tool for hypocalcaemia (ionised calcaemia as the gold standard) among pregnant women in the Nkongsamba Regional Hospital, Cameroon.

### Methods

Our study will target a total of 1067 term pregnant women who shall be included consecutively into the study as they arrive the maternity of the Nkongsamba Regional Hospital for their last antenatal care visit. Data shall be collected using a semi-structured interview-administered questionnaire and blood samples collected for total plasma calcium, albumin

**Data Availability Statement:** All relevant data from this study will be made available upon study completion.

**Funding:** The author(s) received no specific funding for this work.

**Competing interests:** The authors have declared that no competing interests exist.

and serum ionized calcium assays. Additional data will be collected at birth (maternal and foetal variables; foetal outcomes evaluated as secondary outcomes). Total calcaemia and albuminemia shall be measured by atomic absorption spectrophotometry, while ionised calcaemia will be measured by ion-selective electrode potentiometry(using MSLEA15-H electrolyte analyzer) per standard BIOLABO and MSLEA15 protocols, respectively. Data will be analysed using the statistical softwares epi-Info version 7.2.2.16 and STATA version 16.

## Expected research outcome

This study will present a more precise estimate of the burden of hypocalcaemia in late pregnancy as well as identify and analyse the different factors associated with calcium supplementation and ionised hypocalcaemia among term pregnant women in a developing world setting. Maternofoetal outcomes associated with hypocalcaemia will be determined as well as the sensitivity and specificity of total and albumin-corrected calcaemia in diagnosing hypocalcaemia. Our findings will contribute significantly to designing or strengthening interventions to control this electrolyte imbalance.

## Introduction

Hypocalcaemia in pregnancy remains a widespread laboratory and clinical finding in pregnancy [1]. A series of hormonal changes during pregnancy works towards maintaining relatively stable serum calcium levels [2, 3]. Even though multiple studies have reported that ionised serum concentrations in pregnancy and breastfeeding are rarely perturbed [4, 5], these concentrations are generally maintained stable at the detriment of the maternal skeletal stores if adequate replenishment is not encouraged [1, 6].

Two recent systematic reviews [7, 8] agree that women in low and middle-income countries are significantly more likely to have suboptimal calcium intake and therefore, low serum calcium concentrations. According to the World Health Organization (WHO), chronic insufficient calcium intake before or during pregnancy is associated with a significantly increased likelihood of hypocalcaemia in pregnancy [6]. A recent cross-sectional survey evaluating albumin-corrected serum calcium levels in women in late pregnancy reported an alarmingly high prevalence of hypocalcaemia (58%) among women in the Nkongsamba Health District; Cameroon [9]. Similar findings had been reported among pregnant women in the third trimester in Algeria (70%) [10] and India (66%) [11].

This electrolyte imbalance has been significantly associated with hypertensive diseases in pregnancy [12–14] which are alone responsible for more than 64000 maternal deaths annually [15]. Hypocalcaemia in pregnancy has been associated with foetal growth restriction (which alone causes about 40% of stillbirths) [16], neonatal low bone mass [17], increased risk of small for gestational age [18], and increased maternal serum lead levels [19] all with a significant potential of increasing maternofoetal and neonatal morbi-mortality.

In defining hypocalcaemia in pregnancy, authors have used total or albumin corrected values. This could significantly affect the perceived burden of this electrolyte imbalance. The recent study in Cameroon (Nkongsamba Regional Hospital) reported an albumin-corrected hypocalcaemia prevalence in pregnancy [9]. This prevalence is likely to be different when considered from the standpoint of the ionised fraction of serum calcium (the physiologically active fraction). Moreover, very few sensitivity and specificity studies have been carried out to evaluate albumin-corrected calcaemia as a diagnostic test for hypocalcaemic states. In one of such

studies amongst dogs on critical care, albumin-corrected calcium values failed to classify cal-cium status in 67.9% cases accurately. The sensitivity and specificity of total calcaemia in diag-nosing hypocalcemia in this study was 100% and 47%, respectively [20]. The sensitivity and specificity of calcium diagnostic techniques have been evaluated in different pathological states, but gaps of this information persist in pregnancy. It is therefore still very likely that the burden of the imbalance in pregnancy is not accurately estimated. The present study will give estimates of the burden of hypocalcaemia in late pregnancy by measuring ionised calcium lev-els by ion-selective electrode potentiometry (the gold standard of measurement).

Even though highly recommended since 2013, prescription of calcium supplements during pregnancy by the medical personnel remains non-systematic and is judged case by case in our setting. According to a large scale cross-sectional survey on adherence to micronutrient sup-plementation in pregnancy carried out in China, only 57% of pregnant women took any form of calcium supplements during their pregnancies [21]. In this study, adherence levels to cal-cium supplementation were as low as 11.7%. In Africa, a recent survey among pregnant women followed up at the Nkongsamba Regional Hospital reported that 43% of pregnant women went through their pregnancy without any form of calcium supplementation. For the 57% who took calcium supplements, the mean calcium supplementation period in pregnancy was only four months [9]. The mean duration of calcium supplementation in pregnancy in the study by Ajong *et al* was very short; about four months significantly insufficient to meet mater-nofoetal calcium demands.

Very few studies have identified and analysed barriers associated with calcium supplemen-tation in pregnancy. Studies in this domain are even more sparse in the African setting with its particular socio-economic, demographic, and cultural characteristics. A study carried out in Bangladesh, reports calcium supplementation in pregnancy to be dependent on several factors. Maternal knowledge, household factors like high support from husband or partner, reminder from household members to take supplements [22], and health service factors like early initia-tion to prenatal visits, a high number of prenatal visits, and free reception of supplements have been found to affect supplementation in pregnancy [22]. Factors affecting supplementation are expected to vary depending on the socio-economic, demographic, and cultural status of a population. Cameroon is a developing nation with different socio-economic, demographic, and cultural status. In addition, the nutritional habits of this population are very different from that of the Bangladeshi. Assessing the factors influencing calcium supplementation in our setting is therefore indispensable.

Moreover, even if calcium supplementation is respected in some patients, worries persist on its efficacy in solving the problem of hypocalcaemia in pregnancy especially in a zone with feeding habits which have not been fully explored (possible consumption of high levels of cal-cium absorption inhibitors). It is therefore vital to verify if the factors influencing calcium sup-plementation do affect the calcaemic status of these pregnant women. Also, it is essential to know what parameters of calcium supplementation (the type of calcium supplement, the dose of calcium supplements, associated supplements taken with calcium supplements, the duration of calcium supplements, the number of doses taken in a day..etc.) are associated with normo-calcaemia in pregnancy. In addition, knowledge on how the different nutritional combinations affect calcaemic states in pregnant women in our setting is still lacking.

In order to better define the burden of hypocalcaemia in pregnancy, identify its risk factors and correlates, and barriers to calcium supplementation in pregnancy, this study is designed with the following objectives.

1. Identify demographic, socio-economic, and nutritional risk factors of ionised hypocalcae-mia among women in the NRH (Nkongsamba Regional Hospital).

2. Identify barriers to calcium supplementation in pregnancy among women in the NRH.

3. Determine the prevalence of hypertensive disorders in pregnancy among hypocalcaemic pregnant women in the NRH and describe hypocalcaemia-associated maternofoetal outcomes (with foetal outcomes evaluated as secondary outcomes).

4. Evaluate the sensitivity, specificity, positive and negative predictive value of using albumin-corrected calcaemia to define hypocalcaemia in late pregnancy considering ionised hypocalcaemia as a gold standard.

## Materials and methods

### Ethics statement

Written and signed informed consent will be obtained from all participants before inclusion in this study. For participants below 21 years of age, in addition to signing informed assent forms, informed consent will be obtained from their parents or legal representatives. All participants will be free to withdraw from this study at any point in time without any form of penalisation. Ethical clearance for this study has been obtained from the CAMBIN (Cameroon Bioethics Initiative) Ethics Review and Consultancy Committee (ERCC). The ethical clearance reference number is **CBI/452/** ERCC/CAMBIN.

### Study design

This study will be a cross-sectional hospital-based survey targeting apparently healthy pregnant women at the maternity of the Nkongsamba Regional Hospital. Data collection will involve administration of a semi-structured questionnaire, measurement of parameters and collection of blood samples of the participants.

### Setting

Nkongsamba is a city in western Cameroon precisely in the Littoral region of the country. It constitutes most of the Moungo division. The city as censored during the 2005 population censors had a population of 104,050 inhabitants. It is located between two mountains (between Manengouba and Mount Nlonako and about 149 km from the economic capital of Cameroon (Douala). Our study will be carried out at the maternity unit of the Nkongsamba Regional Hospital. This hospital represents the major referral hospital of the Moungo Division. Maternity statistics of this hospital reveal that about 150 new cases of pregnancy are received in this maternity on a monthly. A recent cross-sectional survey in this maternity reports a very high prevalence of hypocalcaemia in late pregnancy (58%) with associated wide gaps in calcium supplementation [9]. This setting is chosen for this study in view of better elucidating calcium imbalance and finding plausible explanations for this problem among pregnant women of this locality.

Collected blood samples shall be analysed at the Moungo branch of the Bethlehem group of laboratories. The Bethlehem group of laboratories constitute a network of multipurpose clinical laboratories of biomedical analyses with over 50 branches scattered over the Cameroon territory. The Moungo branch has a sophisticated biochemistry department with a semi-automatic spectrophotometer. The research team will provide the ion-selective electrode potentiometry apparatus for measurement of ionised calcaemia (MSLEA15-H electrolyte analyzer) using its standard protocols.

## Study duration

Our study is designed to last from September 2020 to December 2021. The detailed calendar of activities is presented below.

## Study population

Our study targets pregnant women accessing antenatal care services at the maternity of the NRH.

**Eligibility criteria.** All pregnant women received at the NRH maternity for routine antenatal care in late pregnancy (greater than or equal to 37 weeks of gestation, given that majority of foetal growth which requires the calcium for foetal bone development occurs in the third trimester and can significantly deplete maternal calcium stores) [23].

**Exclusion criteria.** Our study will however exclude participants with chronic diseases like diabetes, hypertension or known to be carriers of parathyroid disease or vitamin D deficiency (including its potential causes like chronic hepatic disease, chronic kidney disease or malnutrition). Also women on medications known to alter calcaemia like lithium, hydrochlorothiazide, chlorthalidone, phenytoin, prolia, sensipar, cisplatin, bisphosphonates shall be excluded. The exclusion of these participants will be based on consistent verification of their medical records and past medical history, their declarations, clinical evaluation. Participants with intra-uterine foetal demise and multiple gestations will also be excluded.

## Sampling and sample size

Participants shall be consecutively included in the study as they are received at the maternity of the NRH. The minimum required sample size for objective 1 and 2 is calculated using the Cochran's formula for cross-sectional surveys. The expected proportion of women with ionised hypocalcaemia in late pregnancy (considered 50% for maximum sample size, given that no previous data was available), the absolute precision required on either side of the proportion (0.03), and the threshold of error are parameters which are considered by this formula. The minimum sample size of 1067 pregnant women in late pregnancy shall be used.

Concerning objective 3, which evaluates maternofoetal outcomes, the following parameters were considered in evaluating the minimum required sample size. For this, our exposure was hypocalcaemia, and the primary outcome was hypertension in pregnancy. The foetal outcomes will be evaluated as secondary outcomes)

$\alpha$: The Type I error probability for a two-sided test which was set at 0.05.

*power*: The probability of correctly rejecting the null hypothesis was set at 0.80

$p_0$: is the probability of the outcome for a control patient. The prevalence of hypertension in normocalcaemic pregnant women in a recent study was 6.85% (9).

$p_1$: is the probability of the outcome in an experimental subject. The prevalence of hypertension in hypocalcaemic women in a recent study was 14.40% (9).

$m$ is the ratio of control to exposed subjects. That is the ratio of normocalcaemic to hypocalcaemic subjects.

The above parameters were substituted into the power and sample size(PS) software calculator version 3.1 and a sample size of 260 hypocalcaemic and 260 normocalcaemic subjects gotten.

As concerns objective 4, the required sample sizes were obtained from tables for sample size calculation for sensitivity and specificity analysis; for a disease prevalence ranging from 30% to 60% [24]. The prevalence of hypocalcaemia in late pregnancy considered at 60% as was the case in the recent study in NRH [9], the null and alternative hypothesis values of 0.50 and 0.60 respectively, and a p-value of 0.047 were considered. The minimum number of cases for

positive disease were 199 and 299 for sensitivity and specificity evaluation, respectively. The total minimum required sample sizes for evaluation of sensitivity and specificity of 332 and 498 respectively was obtained.

Given the common points between the different objectives, our study will consider an overall sample size of 1067 participants which could be adjusted to meet minimal requirements in specific objectives.

## The procedure of implementation and data collection tools

On completion of the protocol, the questionnaires and other data collection tools shall be prepared. All these will then be pretested on a sample of 50 pregnant women for feasibility and subsequent validation in the Kekem District Hospital, West of Cameroon. Upon validation, administrative authorisations shall be solicited from the director of the Nkongsamba Regional Hospital and the Bethlehem group of laboratories. Six state registered nurses shall be recruited and trained within three training sessions, each session lasting for 5 hours on the objectives of the survey, the consenting process and data collection procedures.

The data for this study shall be obtained by an interviewer-administered semi-structured questionnaire which is going to have different sections (see questionnaire). Additional information shall be gotten by measuring parameters of participants and their corresponding babies, and from blood assays of the pregnant women. The first section of the questionnaire will evaluate the socio-demographic, economic, and obstetric characteristics of the participants. Section 2 will contain specific questions regarding the nutritional behaviour of the participants, while section 3 will be related to calcium supplementation and its co-variables. Section 4 will be a parameter based section in which maternal parameters like height, weight, and blood pressure, as well as foetal parameters, shall be registered. To this section will be added measurements of blood biochemical parameters (total plasma calcaemia, plasma albuminemia, and ionised serum calcium).

## Data collection procedure and blood assays

After a clear explanation of the information sheet to eligible women (taken one at a time), willing participants shall give their written consent to participate by signing an informed consent form (while the consent of legal representatives of minors shall be gotten and they will sign assent forms). In addition to collecting blood samples of participants for subsequent assays, a semi-structured questionnaire shall be administered to each participant by interview. Parameters to measure from each participant shall include the blood pressure, the weight, and the height of the participants. The gestational age of each participant at her booking antenatal (the very first ANC of the woman) visit will be noted in the questionnaire. After about 10 minutes of rest in a sitting position, the brachial blood pressure will be measured using the adapted cuff and an aneroid sphygmomanometer. The cuff shall be placed on the bare skin of the arm, midpoint of the sternum, the arm resting on the table at heart level. Two measurements after a 2minutes interval shall be taken on each arm and the average blood pressure for each arm calculated. Our study will consider the mean blood pressure (BP) from the arm with the higher average [25]. The higher mean BP shall then be registered in to the questionnaire to the nearest 1 mmHg.

The weight of every participant will be measured in a standing position using a digital weighing scale (in kilogrammes), and the height of each participant will be taken in an erect position, using a graduated height measuring scale (in meters). Weight and height of participants will be taken with light clothing, emptied pockets, and shoes off. Bodyweight will be taken to the nearest 0.1 kg and height to the nearest 0.5 cm. Body Mass Index (BMI) in this

study will be calculated as measured weight minus 1 kg (adjusting for clothing), divided by height squared (kg/m$^2$) (9).

According to the International Federation of Clinical Chemistry (IFCC), heparin can be used as the anticoagulant of choice for the determination of ionised calcaemia [26]. Heparin is described to significantly bind calcium in the sample depending on the quantity of heparin used. Measurement of ionized calcium in plasma has proven to give significantly lower concentrations compared to measurement in serum [26, 27]. Given that this survey will determine the burden of hypocalcaemia in this population, we will measure ionized calcium levels in serum.

After allowing the participant to relax and breathe calmly for 10 min, 5ml of venous blood shall be collected using vacuated needles from each consenting participant following WHO best practices in phlebotomy [28] into evacuated lithium heparinised tubes (for measurement of total calcium and albumin in plasma). In addition, 10 ml of blood shall be collected into non-heparinised, dry vacutainer tubes and allowed to stand for 20–30 minutes for serum extraction (the serum will be used for measurement immediately after extraction). During sample collection, tourniquets will be placed for less than 1 min, and the participants would be advised not to exercise the forearm or make a fist during the procedure. All heparinized tubes will be filled to their maximum indicated capacity, thoroughly mixed to distribute the anticoagulant, kept sealed during analysis and handled anaerobically. All collected samples will immediately be transported within 10 minutes to the laboratory and analysed under ambient temperature conditions within the next 20–30 minutes.

The women included in the study shall be seen again at delivery, and additional data on the mother-baby pair collected. These will include the Foetal birthweight (FBW), the brachial circumference (BC), the head circumference (HC), the foetal length (FL), the first and fifth minute Apgar Score (AS)which are going to be evaluated as secondary hypocalcaemia-associated outcomes. The birthweight shall be taken using a digital baby weighing scale and registered to the nearest gramme. The BC, HC, FL are going to be measured using a measuring tape and registered to the nearest centimetre. The AS shall be calculated from evaluation of the five parameters of the APGAR score (scored on a scale of 0–10), at the first and fifth minute of life.

Ionised serum calcium levels are usually measured using ion-specific electrodes and pH-adjusted because the protein binding of calcium is affected by pH. Approximately, half of total serum calcium is in the "free" or ionised state; approximately 40% is bound to serum proteins, principally albumin, and the remainder is bound to anions. Ionised serum calcium is the biologically active fraction of total serum calcium. Total plasma or blood calcium and albumin concentrations will be measured in this study by atomic absorption spectrophotometry. This simultaneous measurement of plasma albumin will permit total calcium levels to be corrected for albumin changes in pregnancy. Variation in plasma albumin concentration alters the concentration of total blood calcium, while the concentration of physiologically important ionised calcium remains unchanged [29]. We will use Payne's equation; $Ca_{adjusted}$(mmol/L) = $Ca_{total}$ (mmol/L) + 0.02 [40 –albumin (g/L)] [30], for the estimation of corrected calcaemia values for each patient. This equation is routinely used in clinical practice to estimate corrected calcium concentrations in patients with hypoalbuminemia. We will adopt this equation for this study, given that pregnant women are generally exposed to reduced serum albumin concentrations.

CPC (O-Cresol Phtalein Complexone) method is widely accepted for the measurement of total Calcium concentration in serum, plasma or urine. The semi-automatic spectrophotometer KENZA MAX of the manufacturer BIOLABO will be used to measure plasma calcium concentrations (using the O-Cresol Phtalein Complexone reagent of BIOLABO) per BIOLABO standard operating procedure [31]. The plasma albumin concentrations will be measured using the same spectrophotometer with bromocresol green reagent of BIOLABO as per the BIOLABO standard operating procedure [32].

We will also directly measure ionised calcaemia and blood pH by ion-selective electrode potentiometry (using the MSLEA15-H blood electrolyte analyser and its standard reagent kits and solutions all manufactured from Guangzhou Medsinglong Medical Equipment Co., Ltd, china) per standard operation procedures. Measured and pH-corrected ionised calcium concentrations shall be reported.

## Data management and data analysis

All completed questionnaires on monthly bases will be sent to the principal investigator for verification and validation. All questionnaires lacking coherence, vital information on the participant or for which well-identified samples are not associated shall be eliminated. All validated questionnaires shall be double entered, compared, and cleaned using the statistical software epi-info version 7.2.2.16. Data shall be imported and analysed in STATA version 16.

Hypocalcaemia defined from total calcaemia will be defined as a total corrected calcaemia of less than 8.5 mg/dl. Women who will have ionised calcium levels less than 4.5 mg/dl will be considered to have ionised hypocalcaemia. Participants with mean systolic blood pressure greater than or equal to 140mmHg and or diastolic blood pressure greater than or equal to 90 mmHg will be considered to have high blood pressure in pregnancy.

Major statistical analyses shall involve the calculation of frequencies and their 95% confidence intervals for categorical variables (prevalence of hypocalcaemia, prevalence of hypertensive disorders in pregnancy, marital status, level of education, occupation) and means/medians for continuous variables (age, calcaemia, blood pressure). For objective number 1, 2, and 3, the strength of association between selected covariates (independent variables: potential predictors and selected outcomes) and hypocalcaemia, calcium supplementation (dependent variables) will be estimated using the odds ratio and its confidence interval at 95% through bivariate analysis. Multiple logistic regression models shall be created and tested for the best fit using known strategies such as testing for collinearity and Archaik information criterion, and subsequent control of potential confounders. All variables showing an association with calcium supplementation and hypocalcaemia with a $p<0.25$ during bivariate analysis shall be included in the multiple logistic regression model for analysis. Our threshold for significance of all the tests shall be set at p-value = 0.05. The evaluation of the accuracy of albumin-corrected calcaemia in defining hypocalcaemia in pregnancy shall be evaluated by estimating the sensitivity, specificity, predictive values of a positive and negative test as well as the likelihood ratios and Cohen's kappa agreement, alongside their 95% confidence intervals. The software STATA version 16 will be used for analysis.

## Expected outcome of the research

At the end of this piece of work aim at throwing more light on serum calcium imbalance, risk factors, calcium supplementation and its barriers. we hope to bring out the following clearly:

✓ State if socio-demographic, economic and nutritional factors influencing the likelihood of hypocalcaemia in pregnancy and go further to describe in what way each factor influences this metabolic imbalance among women in the Nkongsamba Regional Hospital.

✓ Identify factors that influence calcium supplementation in this population and correlate the likelihood of hypocalcaemia to calcium supplementation linked variables (its duration, posology, time of intake, association to other drugs)

✓ Report the prevalence of hypertensive disorders in pregnancy in the hypocalcaemic subpopulation and identify hypocalcaemia-associated maternal and foetal (secondary) outcomes.

✓ Estimate the accuracy of albumin-corrected calcaemia as a tool for diagnosis of hypocalcaemia in pregnancy considering ionised calcaemia as a gold standard.

## Potential impact and scientific implications

Results from this study will present base-line information vital in designing interventions aimed at rolling-back the high prevalence of hypocalcaemia in pregnancy and its associated maternofoetal morbi-mortalities.

Furthermore, an understanding of the factors associated with hypocalcaemia in late pregnancy could serve to control and meet up with this electrolyte imbalance in pregnancy. Comparison of these factors with factors influencing calcium supplementation could help throw more light to the binding nature of hypocalcaemia and calcium supplementation. The contribution of calcium supplementation in maintaining normocalcaemia shall be evaluated by establishing statistical links between effective calcium supplementation and serum calcium levels. Notwithstanding, negative findings will not be surprising. Calcium supplementation in pregnancy might fail to solve the problem of hypocalcaemia in pregnancy because of multiple reasons (nutritional habits, poor prescription habits etc). Most women in Nkongsamba likely go into pregnancy already relatively hypocalcaemic or with low-normal serum calcium levels.

Generally speaking, results of this study will not only help design interventions to fight this electrolyte imbalance in this context but will go a long way in helping prevent low-calcium associated maternal and foetal morbi-mortality in our setting. Also, our research will help generate new hypotheses for further research in this field all to improve mother and child health in Cameroon.

The design adopted in this protocol however has some limits. This study is cross-sectional and will not provide information on causal links. The gestational age at recruitment (>37 weeks) will not allow for the evaluation of some important outcomes like small for gestational age, prematurity, foetal growth restriction, and pregnancy induced hypertension. However, given that the pivot of our study is around calcium and hypocalcaemia, we chose to enroll women late in pregnancy because the great deal of foetal growth occurs in the third trimester. This is likely to cause serious depletion on maternal calcium stores that will carry into the breastfeeding period. Including women earlier than at term is likely to cause us to have an erroneous value of the burden of hypocalcaemia given that the first and second trimester of pregnancy are generally associated with a building of calcium stores awaiting use in the third trimester [33]. It is therefore logical to include these women only when a great deal of growth in the third trimester has occurred so as to possibly evaluate the effectiveness of the stores in meeting maternofoetal demands. Lastly, other important outcomes like neonatal hypocalcaemia will not be evaluated using this protocol and the exclusion of other possible causes (although these causes are rare) of hypocalcaemia according to this protocol is not exhaustive and is not based on laboratory tests.

## Dissemination of research results

The results of this study will be presented at conferences and published in a peer-review journal (PLoS one). The results will guide future population-specific interventions. The final database shall also be deposited into a PLOS one data repository for future use.

## Ethical considerations

### Risks

potential risks associated with this study include the bridge of confidentiality of participants, non-respect of person and autonomy. In addition, the study will involve drawing blood from a vein of the participant. The risks involved in drawing blood from a vein may include, but are

not limited to, momentary discomfort at the site of the blood draw, possible bruising, redness, and swelling around the site, bleeding at the site, feeling of lightheadedness when the blood is drawn, and rarely, an infection or hematoma at the site of the blood draw.

## Benefits

This study will document risk factors of ionised hypocalcaemia in pregnancy and associated maternofoetal outcomes in the Nkongsamba health District; Cameroon. Our findings are vital if not indispensable for effective control and prevention of calcium-associated metabolic imbalancein pregnancy. Participants will not directly benefit from the study, and no financial incentives would be given to participants for their participation. However, women detected with severe hypocalcaemia will be prescribed adequate follow-up.

## Confidentiality/Risks minimization

Several measures shall be taken to minimise the exposition of the participants to the risks mentioned above and to guarantee the confidentiality of participants. First of all, ethical clearance for this study has been obtained from the CAMBIN (Cameroon Bioethics Initiative) Ethics Review and Consultancy Committee (ERCC). Notice of information will be introduced to participants or their legal representatives during which they will be well enlightened on the objectives of the study and the potential risks associated, leaving the participant to willingly accept to participate or withdraw from the study without any compensation for the participants nor any penalisation for those that refuse to participate. If accepted, informed consent will be signed by them before enrollment. Young pregnant women below 21 years of age shall sign assent forms and consent will be obtained from their legal representatives or guardians. Data collected will be anonymous and shall remain confidential between the research team and the team of health personnel involved in the investigation. This will not be disclosed to a third party without the consent of the participants. The questionnaire used for surveys will be treated using codes to avoid the identification of the participants. To minimise risks associated with the blood draws, all blood draws for this purpose shall be done by well-trained nurses following WHO Guidelines on Drawing Blood: Best Practices in Phlebotomy [28]. In addition, any complications following blood draws shall be managed free of charges for the participant in the hospital hosting the study.

## Research plan

The different activities and the time periods are presented on Table 1 below.

**Table 1. Calendar of activities.**

| Period<br>Activity | November 2019-June 2020 | May 2020-June 2020 | June 2020-July 2020 | August 2020-June 2021 | July 2021-August 2021 | September 2021-May 2022 |
|---|---|---|---|---|---|---|
| Development of protocol and data collection tools | ■ | | | | | |
| Administrative Authorisations | | ■ | | | | |
| Pretesting | | | ■ | | | |
| Ethical evaluation and approval, Acquisition of research materials and equipment, Training of data collectors | | | ■ | | | |
| Data collection | | | | ■ | ■ | |
| Data management, entry and Analysis | | | | | ■ | ■ |
| Reporting and publishing of research findings | | | | | | ■ |

## Supporting information

**S1 Data. Research questionnaire /data collection tool.**
(DOCX)

## Author Contributions

**Conceptualization:** Atem Bethel Ajong, Bruno Kenfack, Innocent Mbulli Ali, Martin Ndinakie Yakum, Phelix Bruno Telefo.

**Formal analysis:** Martin Ndinakie Yakum.

**Funding acquisition:** Atem Bethel Ajong.

**Investigation:** Atem Bethel Ajong, Phelix Bruno Telefo.

**Methodology:** Atem Bethel Ajong, Bruno Kenfack, Innocent Mbulli Ali, Martin Ndinakie Yakum, Loai Aljerf, Phelix Bruno Telefo.

**Project administration:** Atem Bethel Ajong, Phelix Bruno Telefo.

**Resources:** Atem Bethel Ajong, Loai Aljerf.

**Software:** Atem Bethel Ajong, Martin Ndinakie Yakum.

**Supervision:** Atem Bethel Ajong, Bruno Kenfack, Innocent Mbulli Ali, Loai Aljerf, Phelix Bruno Telefo.

**Validation:** Atem Bethel Ajong, Bruno Kenfack, Innocent Mbulli Ali, Martin Ndinakie Yakum, Loai Aljerf, Phelix Bruno Telefo.

**Writing – original draft:** Atem Bethel Ajong.

**Writing – review & editing:** Atem Bethel Ajong, Bruno Kenfack, Innocent Mbulli Ali, Martin Ndinakie Yakum, Loai Aljerf, Phelix Bruno Telefo.

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
