## [Decision Letter · Decision Letter 0]

16 Sep 2020

PONE-D-20-23202

Hypocalcaemia and calcium intake in pregnancy: a critical analysis of risk factors, maternofoetal outcomes and evaluation of diagnostic methods in a third-class health facility

PLOS ONE

Dear Dr. Ajong,

Thank you for submitting your manuscript to PLOS ONE. After careful consideration, we feel that it has merit but does not fully meet PLOS ONE’s publication criteria as it currently stands. Therefore, we invite you to submit a revised version of the manuscript that addresses the points raised during the review process.

Four experts in the field handled your manuscript, and we are very thankful for their time and efforts. Although interest was found in your study, several major concerns arose during review that overshadowed this enthusiasm. Notably, the introduction needs to better reflect the rationale for this study; there are questions about the experimental design and endpoints; there are concerns about the methods for measuring blood pressure and the interpretation of these values; further explanation of the statistical analyses needs to be provided; and it is not clear if all of the limitations of this study were addressed in the discussion. Please respond to ALL of the reviewers' comments in your revised manuscript.

We look forward to receiving your revised manuscript.

Kind regards,

Frank T. Spradley

Academic Editor

PLOS ONE

Reviewers' comments:

Reviewer's Responses to Questions

**Comments to the Author**

1. Does the manuscript provide a valid rationale for the proposed study, with clearly identified and justified research questions?

Reviewer #1: Yes

Reviewer #2: Yes

Reviewer #3: Yes

Reviewer #4: Yes

2. Is the protocol technically sound and planned in a manner that will lead to a meaningful outcome and allow testing the stated hypotheses?

Reviewer #1: Yes

Reviewer #2: Partly

Reviewer #3: Partly

Reviewer #4: Yes

3. Is the methodology feasible and described in sufficient detail to allow the work to be replicable?

Reviewer #1: Yes

Reviewer #2: Yes

Reviewer #3: Yes

Reviewer #4: Yes

4. Have the authors described where all data underlying the findings will be made available when the study is complete?

Reviewer #1: Yes

Reviewer #2: Yes

Reviewer #3: Yes

Reviewer #4: Yes

5. Is the manuscript presented in an intelligible fashion and written in standard English?

Reviewer #1: Yes

Reviewer #2: Yes

Reviewer #3: No

Reviewer #4: Yes

6. Review Comments to the Author

You may also provide optional suggestions and comments to authors that they might find helpful in planning their study.

Reviewer #1: This is a very good study and I liked the background of research.

In title you have mentioned about the "Third class health facility" - Rather than using this sentence you can use "Level III health facility" or "third category health facility".

In Introduction part: You have mentioned about taking blood lead level. How will this be helpful in finding out its effect in blood calcium level? What is the evidence that blood calcium level varies with high blood lead level.

It would have been better if you could have separated introductory part with the review of literature.

You also have mentioned about blood copper level in your proforma but has not mentioned its significance in you introductory part. Please include this too if you are collecting data and analyzing it.

You stated that you will use EPI-info-7 for data analysis. Is this software enough to carryout all the analysis you require?

Reviewer #2: Major comments

1. The use of mean blood pressure reading from both arms is not always correct and can either underestimate or overestimate the actual blood pressure. It is only useful if the difference is less than or equal to 10 mmHg. Kindly refer to the American Heart Association guide for blood pressure check.

2. Are you estimating iCa in the serum or plasma? There seem to be some conflicts with this in your methodology. For this study looking at the burden of hypocalcaemia, serum estimation will be more appropriate, and the methodology should support this if that is the case. The use of heparinized bottles overestimates hypocalcaemia even when calcium titrated heparin bottles are used. Not just from the heparin binding the calcium but from other blood components like the erythrocytes. There are several studies in support of this.

3. Change in blood pressure between booking and presentation of 30 mmHg systolic or 15 mmHg diastolic has long been removed from criteria for making diagnosis of hypertensive disorders in pregnancy. This is because there have been no associated adverse effects with this. Check the National High Blood Pressure Education Working Group report.

4. What is unique with a p value of <0.25 as criteria for inclusion in the multiple regression model? While not all variables or the significant ones in the bivariate analysis i.e. p<0.05?

5. Data management and analysis should be presented according to the objectives.

6. Your questionnaire suggest that you are assessing for levels of lead and copper. Include details of your assessment of lead and copper in your methodology and it needs to be appropriately justified in your introduction section.

The minor comments are as attached

Reviewer #3: 1- the keywords are not related to the topic presented in your study

2-as mentioned in your protocol; the benefits of this study to assess risk factors for hypocalcaemia ,however you did not search for these risk factors as( Vitamin D deficiency,Magnesium deficiency. Hypoparathyroidism and pseudohypoparathyroidism......., while in eligibility criteria you excluded some of risk factors ? it is not clear which participants you will include?

If you will exclude all causes of hypocalcaemia I think you will not find enough cases with decreased serum calcium level just because of decreased intake

3- many medications and diseases that can affect calcium level were not mentioned

4- if you will focus on dietary and nutritional factors you will need a better questionnaire that could quantify calcium intake accurately

5-regarding outcome measures:

a-gestational age you choose for women enrollment in your study will not allow you to detect important pregnancy complication s as PIH, preeclampsia , preterm labor, FGR ,SGA…….that mostly have been terminated prior to enrollment

b- Definition of HTN in pregnancy is not accurate: Participants with mean systolic blood pressure greater than or equal to 140mmHg and or diastolic blood pressure greater than or equal to 90 mmHg will be considered to have high blood pressure in pregnancy.

You must confirm with 2 readings (on two occasions at least four hours apart)

c-Neonatal hypocalcaemia is an important outcome

6- the data collection sheet:

a-Specify is the dose of calcium supplementation is the elemental calcium,

b- will you measure Blood copper as mentioned in blood assay and why??

c- include previous pregnancies, interpregnancy interval, duration of previous breastfeeding as all can affect calcium stores

7-total calcaemia, ionized calcemia : please refer to as total calcium, ionized calcium levels

Reviewer #4: This study has multiple purposes. It will present the burden of hypocalcaemia in pregnancy as well as identify and analyse the different factors associated with calcium supplementation in Africa. At the same time, the study evaluated the role of corrected calcium concentrations in the diagnosis of hypocalcemia. This paper is a detailed description of the work on the design.

Because pregnant women are generally exposed to reduced serum albumin concentrations, it is better to calculate the incidence of hypoproteinemia in pregnancy. And the differences between the two diagnostic methods(total serum calcium and corrected calcaemia values ) could be compared.

7. PLOS authors have the option to publish the peer review history of their article (what does this mean?). If published, this will include your full peer review and any attached files.

Reviewer #1: **Yes: **Dhruba Shrestha

Reviewer #2: **Yes: **Collins Ejakhianghe Maximilian Okoror

Reviewer #3: No

Reviewer #4: No

---

## [Author Response · Author response to Decision Letter 0]

22 Sep 2020

Response to reviewers’ comments

Reviewer’s comments and Response to reviewer’s comments

Reviewer number 1

This is a very good study and I liked the background of research.

 Ans: Thank you for the appreciation and the constructive review. 

In title you have mentioned about the "Third class health facility" - Rather than using this sentence you can use "Level III health facility" or "third category health facility". 

 Ans: Thanks for the correction. The modification has been adopted in the title(See title).

In Introduction part: You have mentioned about taking blood lead level. How will this be helpful in finding out its effect in blood calcium level? What is the evidence that blood calcium level varies with high blood lead level.

 Ans: Thank you for the question. Initially the project was supposed to measure lead levels in blood so as to describe the lead poisoning profile in the population but because of limited funds, that section has been suspended. Associations have been established in literature between hypocalcaemia in pregnancy and lead poisoning. The evidence is presented in the introduction, page 4, last paragraph. We have however decided to take off this information and remove anything on lead in the protocol to avoid any confusion because this protocol will not be interested in blood lead levels. 

It would have been better if you could have separated introductory part with the review of literature.

 Ans: Thanks for the suggestion. Originally, we intended to provide a literature review section but from the guidelines for protocols on the journal, we did not see any section destined for literature review. We however tried our best to address all the background issues in the introduction.

You also have mentioned about blood copper level in your proforma but has not mentioned its significance in you introductory part. Please include this too if you are collecting data and analyzing it.

 Ans: Thanks for the suggestion. Our study will not measure copper levels in the blood of the participants. We have deleted this from the questionnaire. See questionnaire, page 20-26

You stated that you will use EPI-info-7 for data analysis. Is this software enough to carryout all the analysis you require? 

 Ans: Thank you for the suggestion. We will use Epi-info and STATA. This has been included in the manuscript. See page 14, last paragraph, line 333-335

Reviewer #2: Major comments

Thank you for asking me to review this protocol titled: “Hypocalcaemia and calcium intake in pregnancy: a critical analysis of risk factors, maternofoetal outcomes and evaluation of diagnostic methods in a third-class health facility”

Hypocalcaemia in pregnancy remains a problem in the LMICs and every measure to tackle it and thus reduce the associated potential adverse maternal and foetal effects will be very helpful. Hence, I consider this study very useful.

I, however, have made some observations and will be worth addressing them by the authors to improve the quality of the study. 

Ans: Thanks very much for your appraisal. Thanks for the constructive review comments.

1. The use of mean blood pressure reading from both arms is not always correct and can either underestimate or overestimate the actual blood pressure. It is only useful if the difference is less than or equal to 10 mmHg. Kindly refer to the American Heart Association guide for blood pressure check.

 Ans: Thanks for the correction. We have verified and corrected this on the manuscript. We will consider the mean BP from the arm with a higher mean BP. See page 12, paragraph 1, line 254-266

2. Are you estimating iCa in the serum or plasma? There seem to be some conflicts with this in your methodology. For this study looking at the burden of hypocalcaemia, serum estimation will be more appropriate, and the methodology should support this if that is the case. The use of heparinized bottles overestimates hypocalcaemia even when calcium titrated heparin bottles are used. Not just from the heparin binding the calcium but from other blood components like the erythrocytes. There are several studies in support of this.

 Ans: Thanks for the pertinent observation. We have provided the changes in the manuscript. We will measure total calcium and albumin in plasma and ionized calcium will be measured in serum. See the methodology, last paragraph of page 12/13, line 275-293.

3. Change in blood pressure between booking and presentation of 30 mmHg systolic or 15 mmHg diastolic has long been removed from criteria for making diagnosis of hypertensive disorders in pregnancy. This is because there have been no associated adverse effects with this. Check the National High Blood Pressure Education Working Group report. 

 Ans: Thanks for the pertinent correction. This has been removed from the criteria in the manuscript. See the methods, page 15, paragraph 1, line 340-342.

4. What is unique with a p value of <0.25 as criteria for inclusion in the multiple regression model? While not all variables or the significant ones in the bivariate analysis i.e. p<0.05?

 Ans: Thanks for the question. We simply considered this to make sure we considered variables which showed some degree of influence. According to Hosmer et al, variables can be included in the multivariate analysis when a p-value < 0.25 is observed in the univariate analysis. In this manner, it is assured that all pertinent and potentially predictive variables are studied. 

Reference: Hosmer DWJr, Lemeshow S, Sturdivant, RX. Applied Logistic Regression, 3rd ed.; John Wiley & Sons: New York, NY, USA, 2013.

5. Data management and analysis should be presented according to the objectives.

 Ans: Thanks for the suggestion. The data analyses section has been edited accordingly with more clarifications. See data analysis section. Page 14 (last paragraph) and 15.

6. Your questionnaire suggest that you are assessing for levels of lead and copper. Include details of your assessment of lead and copper in your methodology and it needs to be appropriately justified in your introduction section.

 Ans: Thanks for your observation. This is not going to be evaluated in this study. They have been taken off the questionnaire. See the questionnaire

Minor comments

1. “Term” begins at 37 completed weeks. The use of “near term” while your population comprise of women of gestational age 37 weeks and above will not be correct. 

 Ans: Thanks for the correction. This has been corrected in the manuscript. See the abstract, (results section and expected outcome)

2. The terms “structured” and “semi-structured” questionnaire was used interchangeably multiple times in this writing. Looking at your questionnaire, I will advise you stick with “semi-structured” questionnaire. 

 Ans:Thanks for the correction. The term semi-structured questionnaire has been adopted in the manuscript. See method section, page 8, line 155 and page 11, line 235

3. The term “retarded” is an obsolete word to describe foetal growth. We talk of “foetal growth restriction” and not “foetal growth retardation”, that appeared severally in your writing. 

 Ans: Thanks for the correction. Foetal growth restriction has been adopted in the manuscript. See introduction, page 4, line 71 

4. Kindly re-write the first sentence of paragraph 3 of the introduction section as it is not clear in its current form. 

 Ans: Thanks for the observation. The sentence has been modified. See introduction, page 4, paragraph 3

5. While your topic will suggest that you aim to determine the materno-foetal outcomes of hypocalcaemia, your objective 3 suggest that you aim to determine the materno-foetal outcomes of hypertensive disorders in pregnancy. Kindly harmonise and rewrite. 

 Ans: Thanks for the observation. Objective three has been made clear. See objectives, End of page 6

6. Provide reference for Page 8, In. 160-162 

 Ans: Thanks. The reference has been added. See page 8, line 167.

7. The protocol will benefit from a review for coherence and conjunctions 

 Ans:Thanks for the suggestion. The whole write-up has been read and corrections made.

8. On your questionnaire, I have the following observations and comments:

a. I am not sure if the aim of Q10 was to assess parity. If it is, the number of deliveries will be more appropriate rather than number of children alive.

b. The purpose of asking for abortions (better term “miscarriages”) in Q13 is not quite clear from your protocol. However, I will suggest separating miscarriages from stillbirths/ENND.

c. Create opportunity for “others” in Q16 and Q33.

d. Separate iron and folic acid in Q34 as a woman may not be taking both.

e. Question 46 on your questionnaire is same as Q9. I will suggest including the gestational age at recruitment and at delivery as these are potential confounders.

 Ans: Thanks for your suggestions. All these suggestions have been integrated into the questionnaire. See the edited questionnaire

Reviewer 3

1- the keywords are not related to the topic presented in your study

 Ans: Thanks for your observation. We have updated the list of keywords. See lines 48-50

2-as mentioned in your protocol; the benefits of this study to assess risk factors for hypocalcaemia ,however you did not search for these risk factors as( Vitamin D deficiency,Magnesium deficiency. Hypoparathyroidism and pseudohypoparathyroidism......., while in eligibility criteria you excluded some of risk factors ? it is not clear which participants you will include? If you will exclude all causes of hypocalcaemia I think you will not find enough cases with decreased serum calcium level just because of decreased intake 

Ans: Thank you for your observation. Our study is interested in evaluating demographic, socio-economic, and nutritional risk factors for hypocalcaemia in pregnancy. Our major outcome variable being hypocalcaemia, we tried to exclude the most common and evident causes of hypocalcaemia in this section. We excluded patients with known vitamin D deficiencies (identifying all its major causes), carriers of parathyroid diseases which are the two major causes of hypocalcaemia. Other causes are rare, could not be easily excluded without a load of work-ups. We shall however include this as a limit of the study. See methods section, page 9, 187-194 and page 18, 398-407

3- many medications and diseases that can affect calcium level were not mentioned 

Ans:Thanks for your observation. As stated above, we tried to eliminate the major and most frequent causes of hypocalcaemia. We have included drugs which could affect calcium levels in the criteria. See page 9, 187-194

4- if you will focus on dietary and nutritional factors you will need a better questionnaire that could quantify calcium intake accurately 

Ans: Thanks for the suggestion, we are interested on how nutritional habits of participants could play on their blood calcium levels. We do not intend to quantify daily calcium intake in these women given that a lot of recent studies already indicate that in our developing setting, intake is suboptimal.

5-regarding outcome measures:

a-gestational age you choose for women enrollment in your study will not allow you to detect important pregnancy complication s as PIH, preeclampsia , preterm labor, FGR ,SGA…….that mostly have been terminated prior to enrollment 

Ans: Thank you for the observation. We recognize that our study will not be able to evaluate these outcomes. Reason for which we did not present any of these as an outcome. We will present information on hypertensive disease in pregnancy as a block but not pre-eclampsia and PIH. 

Given that the pivot of our study is around calcium and hypocalcaemia, we chose to enroll women late in pregnancy because the great deal of foetal growth occurs in the third trimester. This is likely to cause serious depletion on maternal calcium stores that she will carry into breastfeeding. This will be integrated as a limit of this protocol. See page 18, line 398-407.

b- Definition of HTN in pregnancy is not accurate: Participants with mean systolic blood pressure greater than or equal to 140mmHg and or diastolic blood pressure greater than or equal to 90 mmHg will be considered to have high blood pressure in pregnancy. You must confirm with 2 readings (on two occasions at least four hours apart) 

Ans: Thanks for your correction. The definition given in our manuscript is operational for the study and for statistical analysis. More precisions on the measurement of BP have been given as requested by reviewer 2.

c-Neonatal hypocalcaemia is an important outcome

 Ans: Thanks for the suggestion. We agree but the given the number of outcomes already under consideration and the scope of the objectives of this study. We will prefer to integrate this into a different study. See page 18, line 398-407

6- the data collection sheet:

a-Specify is the dose of calcium supplementation is the elemental calcium,

b- will you measure Blood copper as mentioned in blood assay and why??

c- include previous pregnancies, interpregnancy interval, duration of previous breastfeeding as all can affect calcium stores

Ans:

 a) The doses are elemental calcium doses (specified on questionnaire)

b) Blood copper and lead will not be measured.

c) All these recommendations have been added on the questionnaire

7-total calcaemia, ionized calcemia : please refer to as total calcium, ionized calcium levels Ans: These changes have been made to the questionnaire

Reviewer #4

This study has multiple purposes. It will present the burden of hypocalcaemia in pregnancy as well as identify and analyse the different factors associated with calcium supplementation in Africa. At the same time, the study evaluated the role of corrected calcium concentrations in the diagnosis of hypocalcemia. This paper is a detailed description of the work on the design.

Because pregnant women are generally exposed to reduced serum albumin concentrations, it is better to calculate the incidence of hypoproteinemia in pregnancy. And the differences between the two diagnostic methods(total serum calcium and corrected calcaemia values ) could be compared.

 Ans: Thanks for your observations. This calculations will be integrated when answering objective 4.

---

## [Decision Letter · Decision Letter 1]

15 Oct 2020

PONE-D-20-23202R1

Hypocalcaemia and calcium intake in pregnancy: a research protocol for critical analysis of risk factors, maternofoetal outcomes and evaluation of diagnostic methods in a third-category health facility, Cameroon

PLOS ONE

Dear Dr. Ajong,

Thank you for submitting your manuscript to PLOS ONE. After careful consideration, we feel that it has merit but does not fully meet PLOS ONE’s publication criteria as it currently stands. Therefore, we invite you to submit a revised version of the manuscript that addresses the points raised during the review process.

There are still comments that must be addressed.

We look forward to receiving your revised manuscript.

Kind regards,

Frank T. Spradley

Academic Editor

PLOS ONE

Reviewers' comments:

Reviewer's Responses to Questions

**Comments to the Author**

1. Does the manuscript provide a valid rationale for the proposed study, with clearly identified and justified research questions?

Reviewer #1: Yes

Reviewer #2: Yes

Reviewer #3: Partly

2. Is the protocol technically sound and planned in a manner that will lead to a meaningful outcome and allow testing the stated hypotheses?

Reviewer #1: Yes

Reviewer #2: Yes

Reviewer #3: Partly

3. Is the methodology feasible and described in sufficient detail to allow the work to be replicable?

Reviewer #1: Yes

Reviewer #2: Yes

Reviewer #3: Yes

4. Have the authors described where all data underlying the findings will be made available when the study is complete?

Reviewer #1: Yes

Reviewer #2: Yes

Reviewer #3: Yes

5. Is the manuscript presented in an intelligible fashion and written in standard English?

Reviewer #1: Yes

Reviewer #2: Yes

Reviewer #3: Yes

6. Review Comments to the Author

You may also provide optional suggestions and comments to authors that they might find helpful in planning their study.

Reviewer #1: Thank you so much for the prompt response.

I think the author has made significant changes in the manuscript and now looks good for publication.

I don't have any other issues regarding the article and some of the other major issues that I had thought about has been raised by other reviewers.

Thank you

Reviewer #2: Hypocalcaemia in pregnancy remains a problem in the LMICs and every measure to tackle it and thus reduce the associated potential adverse maternal and fetal effects will be very helpful. Hence, I consider this study very useful. My concerns after my initial review have been addressed by the authors.

Reviewer #3: Thank you for adressing most of my suggestions,however the main issue in methdology is still not answered

The title of your research include fetomaternal outcomes however, you excluded most of these outcomes from being looked for,

I think you can either change your study to just assess risk factors of hypocalcemia and assessing its prevelance without searching for the outcomes

Or you can include all pregnant women in the 3rd trimster admitted for labor and assess for hypocalcemia risk factor , calcium level, obtain your questionnaire that assess different habits and nutritional factors

If you choose the 2nd option I think you have to adjust the sample size to get statistically adequate results.

7. PLOS authors have the option to publish the peer review history of their article (what does this mean?). If published, this will include your full peer review and any attached files.

Reviewer #1: **Yes: **Dhruba Shrestha

Reviewer #2: **Yes: **Collins Ejakhianghe Maximilian Okoror

Reviewer #3: No

---

## [Author Response · Author response to Decision Letter 1]

17 Oct 2020

RESPONSE TO REVIEWERS

We want to appreciate all the reviewers for their time sacrificed to raise very vital comments to better this protocol. The additional comment raised by reviewer three is addressed here.

Comment: Thank you for adressing most of my suggestions, however the main issue in methdology is still not answered. The title of your research include fetomaternal outcomes however, you excluded most of these outcomes from being looked for, I think you can either change your study to just assess risk factors of hypocalcemia and assessing its prevelance without searching for the outcomes Or you can include all pregnant women in the 3rd trimster admitted for labor and assess for hypocalcemia risk factor , calcium level, obtain your questionnaire that assess different habits and nutritional factors. If you choose the 2nd option I think you have to adjust the sample size to get statistically adequate results.

Response: We have a problem considering to recruit and measure calcaemia and other outcomes like BP earlier because;

1. Most often the first trimester and most of the second trimester is dedicated to improvement on the calcium stores of the mother in order to prepare for the high demand in the third trimester. We therefore wanted to measure these calcium levels only when we are sure that a great deal of foetal growth has occurred in the third trimester (1). This will help us evaluate to know whether the calcium stores the women get can really support them for their pregnancy till term (reason for which we considered term pregnancies). If we recruit earlier we will miss a great deal of women who will be susceptible to hypocalcaemia.

2. We could not consider collecting this data in labor and including every woman who presents in labor because, the hyperventilation and strain of the arms and uterine contractions are likely to affect serum calcium levels. In addition, measuring and getting adequate blood pressures in labor will be difficult and contrary to recommendations of standard BP measurement (2)(3). 

Our current design can determine a major maternal outcome which is hypertensive disorder in pregnancy (from which a sample size calculation was done). We recognize that our inclusion gestational age will not allow us measure important foetal outcomes. We have included this as a limit to this protocol (page 16, first paragraph) and have considered to still evaluate these foetal outcomes but precise that they will be evaluated as secondary outcomes. We belief it will be a great waste if we have these babies at term and cannot find out if some of their outcome variables could be linked to hypocalcaemic states of the mother. We have difficulties to shift this time of recruitment of participants because of the two major reasons stated above. 

 References

1. Kovacs CS. Maternal mineral and bone metabolism during pregnancy, lactation, and post-weaning recovery. Physiol Rev. 2016;96(2):449–547. 

2. Muntner P, Shimbo D, Carey RM, Charleston JB, Gaillard T, Misra S, et al. Measurement of blood pressure in humans: A scientific statement from the american heart association. Hypertension. 2019;73(5):E35–66. 

3. Jafri L, Khan AH, Azeem S. Ionized calcium measurement in serum and plasma by ion selective electrodes: Comparison of measured and calculated parameters. Indian J Clin Biochem. 2014;29(3):327–32.

---

## [Editor Report · Decision Letter 2]

21 Oct 2020

Hypocalcaemia and calcium intake in pregnancy: a research protocol for critical analysis of risk factors, maternofoetal outcomes and evaluation of diagnostic methods in a third-category health facility, Cameroon

PONE-D-20-23202R2

Dear Dr. Ajong,

We’re pleased to inform you that your manuscript has been judged scientifically suitable for publication and will be formally accepted for publication once it meets all outstanding technical requirements.

Kind regards,

Frank T. Spradley

Academic Editor

PLOS ONE

---

## [Editor Report · Acceptance letter]

27 Oct 2020

PONE-D-20-23202R2 

Hypocalcaemia and calcium intake in pregnancy: a research protocol for critical analysis of risk factors, maternofoetal outcomes and evaluation of diagnostic methods in a third-category health facility, Cameroon 

Dear Dr. Ajong:

I'm pleased to inform you that your manuscript has been deemed suitable for publication in PLOS ONE. Congratulations! Your manuscript is now with our production department. 

Kind regards, 

on behalf of

Dr. Frank T. Spradley 

Academic Editor

PLOS ONE